# Process Evaluation of a Comprehensive Intervention for the Early Detection and Prevention of Oral Cancer: A Mixed Methods Study

**DOI:** 10.3390/ijerph19127120

**Published:** 2022-06-10

**Authors:** Ibtisam Moafa, Mohammed Jafer, Bart Van Den Borne, Ciska Hoving

**Affiliations:** 1Dental Public Health Division, College of Dentistry, Jazan University, Jizan 45142, Saudi Arabia; majafar@jazanu.edu.sa; 2Department of Health Promotion, Care and Public Health Research Institute, Maastricht University, 6229 HA Maastricht, The Netherlands; b.vdborne@maastrichtuniversity.nl (B.V.D.B.); c.hoving@maastrichtuniversity.nl (C.H.)

**Keywords:** process evaluation, behavior change, early detection, oral cancer, evidence-based practice

## Abstract

Process evaluations help to understand and refine health interventions. The “ISAC” intervention targeted the enhancement of early detection and prevention of oral cancer (OC) through tobacco-cessation counseling, patient communications, and ISAC role-modeling. Over six months, throughout ISAC implementation in the Jazan region of Saudi Arabia, Linnan and Steckler’s process evaluation framework’s specified indicators were assessed, by mixed methods, on context, reach, dose delivered, dose received, fidelity, recruitment, and participant’s satisfaction. Findings showed that 47 of 80 (58.75%) eligible dentists were reached and received all components. Thirty-six (76.6%) participants reported reading all intervention materials, visited the ISAC website, scored high on the perceived quality of provided information (M = 4.62 ± 0.63), and provided support (M = 4.67 ± 0.57). The fidelity was scored high across all intervention components. Role-modeling of the ISAC had the highest satisfaction score (M = 9.77 ± 0.58 out of 10). High perceived-effects were reported in relation to feeling confident in performing OC examination and training patients on OC self-examination (3.95 ± 0.22). The intervention attained high implementation levels for dose delivered, dose received, and fidelity. The intervention delivery was associated with high satisfaction and perceived effects. Using multiple data sources enhanced the understanding of the implementation process and strengthened the validity of the study’s findings.

## 1. Introduction

Oral cancer (OC) is the 16th most prevalent cancer worldwide [1]. OC is a malignant neoplasm that arises in the oral cavity and surrounding tissue, such as the tongue, floor of the mouth, and lips [1,2]. OC commonly originates in the squamous cells and hence 90% of OCs are defined as squamous cell carcinoma [2]. The annual incidence of OC around the world is high: >350,000 cases, with approximately 145,000 deaths [2]. Saudi Arabia has a high incidence rate of OC, of which more than 35% of cases are reported in the Jazan region (south of Saudi Arabia) [2,3]. OC risk increases with age and commonly affects people over 50 years of age [3]. However, there is an increasing trend of OC affecting people of both genders and at a younger age [2]. Globally, the OC age-standardized rate (ASR) is 5.2 for males and 2.3 for females [3]. In Saudi Arabia, ASR for males was 1.4 and 1.6 for females, which contrasts with the global ASR [3]. OC cases in Saudi Arabia, particularly in the Jazan region, were found to be associated with a higher rate of consumption of a local type of smokeless tobacco called Shammah [4]. Shammah is made of ground tobacco, lime, black pepper, and other flavorings [4]. Shammah is placed for a few minutes between the cheeks and teeth, or between the tongue and the floor of the mouth, and is then spat out. Other well-documented major risk factors for OC include alcohol use and human papilloma virus (HPV) [1,2].

OC is a rapidly progressing disease and many of its cases are identified at a late stage; therefore, it is characterized by a poor prognosis and complex treatments [2]. Thus, early detection and prevention of OC is key for a better prognosis and quality of life [2,5]. In fact, previously collected evidence shows that OC screening decreases the mortality of high-risk patients [5]. During OC screening, dentists carefully examine inside the patient’s mouth, lips and throat, and look for red/white patches, mouth sores, lumps or enlarged lymph nodes. Dentists are the most qualified professionals for the early detection and prevention of OC [5]. In previous in-depth investigations of the OC problem in the Jazan region, we found that dentists had a passive attitude towards OC examination and patient education of its risk factors [6]. Moreover, dentists often lack the confidence and necessary skills to perform a complete OC examination and properly administer patient education [6,7]. This was further elicited by reports from dental patients in which they highlighted the passive behavior of their dentists toward OC examination or reported that they had never received education or training in OC self-examination [8].

Therefore, to bridge the gaps observed in the current practice of OC examination and patient education, the ISAC intervention was developed [9,10]. This development was based on a conceptual intervention mapping approach, integrating current evidence, as well as the personal experiences and views of dentists, dental patients, and other relevant community stakeholders [9]. ISAC stands for: I = informing dental patients about performing routine oral cancer screenings, S = screening for oral cancer, A = advising patients at high risk to quit risk factors using clear and tailored language to deliver health messages, and C = connecting patients to specialized centers. The intervention was implemented via theoretical and practical sessions and utilized blended learning. It involved three components: tobacco-cessation counseling, patient-communication skills, and oral cancer screening role-modeling. 

Process evaluation provides important aspects for comprehending and modifying evidence-based intervention. Various theoretical frameworks exist to guide a researcher in the evaluation and implementation process of health interventions [11,12]. In our study, we used a process evaluation framework, developed by Linnan and Steckler (2002), to guide the evaluation process [11]. Therefore, the objective of the present study was to evaluate the implementation process of a multi-component intervention for dentists in order to increase the early detection and prevention of oral cancer by dental interns in areas with a high risk of oral cancer. 

## 2. Materials and Methods

### 2.1. Design

The study made use of a mixed methodology (both qualitative and quantitative data collection) to gather data about Linnan and Steckler’s framework components: context, reach, dose delivered, dose received, fidelity, recruitment, and satisfaction. Data were gathered using a checklist, questionnaire, semi-structured interview, focus group discussion, an attendance log sheet, and the ISAC website analytics. All instruments were pre-tested during the initial phase of the intervention and were adjusted according to the necessary changes [13].

### 2.2. Setting and Participants

Jazan Dental School is the main and largest dental polyclinic in the region, with approximately 80 graduates annually and a high patient-load from different areas. The majority of JDS graduates practice in most of the public and private dental clinics in the Jazan region. Dental interns were targeted in the current intervention because, in the current educational system, they are considered practicing dentists, since they practice clinically, independently, and without clinical requirements or supervision from senior dentists, as was the case in the previous years of studying dentistry. 

### 2.3. The ISAC Intervention 

The ISAC intervention was performed over a 6-month period (October 2020–April 2021) in Jazan Dental School (JDS) clinics during working hours, using both didactic and participatory approaches (PowerPoint presentation, videos, role-play, and modeling). The ISAC protocol was described extensively in Jafer et al. [9] (Appendix A). The intervention is summarized in Table 1 and below. The ISAC consists of three sessions, each addressing its own theme: (1) tobacco-cessation counselling, (2) patient communication, and (3) ISAC modeling. The first session in the intervention is focused on educating and training dental interns on tobacco-cessation counseling, using the WHO’s guidelines for brief tobacco-cessation counseling [14]. The second session is aimed at enhancing the dental interns’ patient-communication skills, making use of an interactive role modeling approach [15]. Finally, the third session looks into presenting and role-modeling the ISAC intervention, integrating the knowledge and skills learned from previous sessions, and providing training on other skills such as OC self-examination for patients and patient referral to specialized centers. Full OC screening followed the oral-cancer clinical- practice guidelines, including, but not limited to, taking patient history (general history and oral history), performing visual examination of extraoral tissues (head, neck, lymph nodes, lips, and perioral tissues), performing intraoral examination (inspect and palpate soft tissues with more attention to high risk areas such as lateral and ventral surface of the tongue, floor of the mouth, and soft palate), evaluating the characteristic of the lesion (color, size, texture, and outline), and the use of adjunctive techniques such as toluidine blue staining and direct fluorescence visualization, as well as taking tissue biopsy [16]. 

### 2.4. Data Collection for the Process Components

We assessed seven outcome measures: the context, reach, dose delivered, dose received, fidelity, and recruitment [11]. Furthermore, we collected data regarding additional process evaluation variables, including the satisfaction, perceived benefits, perceived difficulty, perceived effect of the ISAC, and the quality of the information provided and support during the intervention [12].

In addition to the quantitative satisfaction items, a focus group discussion (FGD) was conducted at the end of the intervention and was moderated by the research team coordinator, who is specialized in dental public-health and has experience in conducting FGDs. Dental interns were informed about the objective of the FGD and were asked to mention and describe the intervention activities that they had enjoyed and benefited from, what insights and knowledge they had gained, and to give feedback on what could have been done differently if the intervention were to be implemented again. The FGD included 20 dental interns of both genders, who were randomly invited to participate. The session lasted for 45 minutes. In addition to the FGD, an individual face-to-face interview was conducted with the ISAC adopters (the clinical director and intern supervisor) and the implementer (a specialist dentist in the dental public health division) to discuss their experiences with the ISAC implementation, challenges they have faced, successful activities, potential areas for improvement, and their intentions towards continually adopting the intervention. 

Context is the physical, political, and social environments that could directly and indirectly influence the intervention [11]. We assessed the context by interviewing the dental interns, as well as by having the observer watch the implementers’ delivery of the ISAC and dental interns’ practice of the ISAC and writing down field-notes about any contextual changes. At the six-month follow-up, the dental interns were asked whether they had accessed other information outside the ISAC intervention and if there have been exposed to any other types of intervention, in order to assess whether the ISAC outcomes were contaminated. 

Reach is the proportion of the target audience who participated in the intervention [11]. We assessed reach by calculating the percentage of eligible dental interns who enrolled in the intervention, the percentage of enrolled dental interns who received the intervention and the percentage of the dental interns who received all intervention components. 

Dose delivered is the amount of the intervention that was delivered to the participants [11]. We assessed it by the number of intervention components delivered to the dental interns.

Dose received is the extent of intervention content that was received by dental interns, calculated as the number of completed sessions and how thoroughly the interns had read the ISAC resources on a scale from 0 (I did not read the ISAC resources: clinical protocol, guidelines for tissue biopsy taking, guidelines for dentist–patient communication, or guidelines for brief tobacco-cessation counseling) to 4 (I have read everything) [11].

Fidelity is the extent to which the ISAC was implemented and delivered as intended [11]. We assessed the fidelity using a 29-item structured observation checklist that was previously published and developed through an iterative process and included a systematic review of evidence, feedback from experts who evaluated the checklist regarding a list of desirable elements, consultations with patients and dentists, and pilot testing of the checklist. Five out of 29 items were specified as OC risk factor counseling components, and included tobacco history, assessing readiness to quit, and encouraging patient to quit (items 7–10, 21). Six out of 29 items specifically pertained to patient communication, and included using clear and simple language to communicate the information the patient wants to know, showing politeness and respect to the patient, and supporting the patient to ask questions (items 22–27) (<ObservationChecklist>, https://osf.io/qsbcu/ accessed on 1 September 2020).

Recruitment concerns the procedure that was used to approach the ISAC participants [11]. Recruitment was assessed by asking the ISAC participants (dental interns, implementers, and adopters) about the procedures used to attract them to participate in the ISAC. 

Participants’ satisfaction with each intervention component was assessed at the six-month follow-up on a 10-point scale (0 = not satisfied at all to 10 = extremely satisfied). 

Perceived benefits: The dental interns rated the extent to which each of the following was beneficial to them: each of the three intervention sessions, the ISAC website, the ISAC modeling video, and the ISAC reading materials from 0 to 10 (not valuable at all to extremely valuable). 

Perceived difficulty: The dental interns rated how difficult each of the following items were for them on a scale from 0 to 10 (not at all difficult to extremely difficult): each of the three intervention sessions, using the ISAC website, understanding the ISAC modeling video, and the ISAC reading materials.

Perceived effect of the ISAC: dental interns rated the degree to which the ISAC changed many aspects of their behavior (greater understanding of oral cancer, improved confidence, improved tobacco counseling, patient education and communication skills, improved biopsy-taking skills, improved OC-examination skills, improved knowledge, and more skills for referring patients to specialized centers) from 0 to 4 (strongly disagree to strongly agree). 

The quality of the information and support provided in general during the intervention was rated by the participants on a scale from 1 to 5 (poor to excellent).

### 2.5. Data Analysis

The reach, dose delivered, dose received, recruitment, participant satisfaction, and contamination were reported using descriptive statistics (frequency, percentage, mean, and standard deviation) of IBMSPSS statistics, version 26. Qualitative data from the participants’ perceived satisfaction, perceived benefits, and facilitators were analyzed manually using thematic analysis [17].

## 3. Results

### 3.1. Reach

Of the 80 dental interns, 47 (58.75%) were identified as eligible, and had consented and enrolled in the intervention (Figure 1). All participants (100%, *n =* 47) completed the baseline OC examination practice measurement and baseline questionnaire and received all intervention components. Of the enrolled dental interns, 85% completed the second measurement at two-months post-intervention and 83% completed the six-month follow-up measurement. More females (53%) than males (46%) were recruited in the intervention at baseline, at two-months post-intervention (females = 60%, males = 40%) as well as at the six-month follow-up (females = 53%, males = 46%). Of the 47 enrolled interns, 41 (87%) had returned the post-intervention questionnaire (females = 48.7%, males = 51.2%). The mean of the ISAC clinical practice time was 9 ± 4 min; (range: 5 to 13 min) (Table 2).

### 3.2. Dose Delivered

Each of the 47 enrolled dental interns engaged in all the intervention components. The component focusing on education and training in tobacco-cessation counseling took five days, with four hours per day. The interactive training on patient communication took three hours. The ISAC role-modelling component included an hour for a theoretical presentation with questions and answers, followed by a practical workshop (modeling and tailored feedback), which took six hours over four successive days, with 11 to 12 dental interns per day. 

### 3.3. Dose Received

Most participants (76.6%) reported reading all the ISAC materials thoroughly. The remaining participants reported not thoroughly reading the resources because of the belief that they already had enough information on OC examination, patient education, and communication (*n =* 5) or lacked enthusiasm for reading in general (*n =* 4). All the enrolled dental interns accessed the ISAC website and watched videos of ISAC role-modeling by their colleague and of patient-communications skills and taking intra-oral biopsies. 

### 3.4. Recruitment

The dental interns were recruited through convenient sampling to include all practicing dental interns of both genders in JDS. The adopters and implementers were recruited via email invitation, and this was followed by face-to-face contact. 

### 3.5. Fidelity

It was confirmed that the fidelity to the intervention manual was high across all three intervention components. All the intervention sessions were implemented according to the ISAC protocol, and the 29 items were checked across all four sessions of the ISAC role-modeling. Five out of the 29 items, which were specified for the tobacco counseling component, were checked across all five sessions. Six out of the 29 items that were specific to the patient-communication component were also all checked in the session.

### 3.6. Satisfaction

The dental interns gave a high rating for their perceptions of their degree of benefit from the ISAC website (M = 9.87 ± 0.34), the ISAC modeling video (M = 9.90 ± 0.30), the ISAC reading resources (M = 8.54 ± 1.68), the patient-communication session (M = 9.95 ± 0.22), and the tobacco counseling session (M = 9.92 ± 0.27), as well as the ISAC modeling session (M = 9.95 ± 0.22). 

Qualitatively, the dental interns reported several benefits from practicing the ISAC intervention, including: reinforcing their oral cancer epidemiology and prevention-related knowledge, recognizing their important role in oral cancer prevention, and improving their confidence and tobacco-counseling skills. When asked in the survey about the greatest benefits of the intervention, the dental interns identified: a better understanding of their role in preventing and detecting oral cancer as well as feeling more confident in their ability to perform oral-cancer examinations and in educating as well as on training patients on how to perform oral cancer self-examinations (Table 3). In addition, the total number of reported potentially malignant oral disorders by dental interns highly increased from zero at baseline to 69 at two months post-intervention and to 85 at follow-up. 

All participating dental interns who completed the follow-up (*n =* 39) reported that they would recommend the ISAC to their colleagues. 


*“I have already recommended the ISAC to all my colleagues, as they have seen how the ISAC has greatly improved my confidence and skills”.*


### 3.7. Context

Due to the global outbreak of COVID-19, there were strict measurements imposed on a national level as well as on a clinical level, which limited the number of potential participants from 80 to 47, and reduced clinical working hours and the number of operating clinics. However, there was no conflict between these changes and the implementation plan of the ISAC protocol. External sources of information included internet sites (OC patient stories on YouTube and Saudi Ministry of Health for additional tobacco awareness brochures) and five participants accessed information from other colleagues (a type of referral system at different clinics). Across the qualitative findings, we identified contextual factors that may influence dentists’ engagement/adherence to the intervention. During the intervention, participants also mentioned that they hoped all dental organizations would focus on implementing OC examinations. 


*“I hope all dental clinics focus on implementing oral-cancer examinations as much as they care about their financial profits”.*


Other aspects related to integrating the intervention checklist into the clinical software. 


*“I think the ISAC would be easier to follow by all dentists if the ISAC was integrated into the clinical software”.*


Other participants preferred to modify the ISAC intervention to support dentists who are practicing without an assistant. 


*“If I practice in a rural area, where I have no assistant, I think it will be challenging for me. Maybe it would be useful if the ISAC was somehow modified to account for this”, “…I would prefer to have a supporting tool that can help me when I practice alone”.*


Some participants believed implementing the intervention at a large scale would influence the adherence to the ISAC. 


*“If it’s implemented at a national level, definitely everybody would practice it”.*


Other participants requested that the intervention be integrated into dental organizations as part of continuous education. 


*“We really need the ISAC to be part of the university’s continuous education course”, “we hope that Jazan University and the Ministry of Health continuously offer this intervention to every dentist in Saudi Arabia”.*


A general summary of the main findings from the ISAC process evaluation is described in Attached Table 4.

## 4. Discussion

This process evaluation provides insights into the implementation of the ISAC intervention in dental clinics. We observed that intervention attendance was high, and all participants of both genders received the three ISAC components. All participants engaged in the three intervention components and most of participants made use of the ISAC resources. Participants reported high satisfaction with the ISAC, seeing many benefits and experiencing little difficulty in applying the OC screening and patient-communication methods. As intended, participants reported that the ISAC increased their awareness of the general problem of OC and their role in prevention and early detection as well as their ability to engage in complete OC examinations and patient-education practices. 

The ISAC website and the ISAC-modeling video were perceived as highly beneficial by the participants and the effect of using the blended learning in health professional education is well documented in the literature [18,19]. It can maximize the intervention effects, without losing the socialization and the collaborative features of face-to-face learning [20]. The utilization of blended pedagogy is commonly practiced within the continuous professional-education context [20]. Through blended learning, health professionals are able to overcome high travel costs, take the courses at their convenience, develop themselves in line with their needs and interests, and mitigate the spread of infection during the pandemic with the outbreak of COVID-19 [20]. Therefore, following a balanced approach between face-to-face and online learning can be highly beneficial in health-promotion and disease-prevention interventions. In fact, it might be the method of choice, particularly during the pandemic, to control the outbreak of infectious diseases [21].

Our findings showed that inadequate work resources are a potential challenge to dentists’ engagement in practicing the ISAC intervention. Work resources are those physical, social, and organizational aspects that reduce work demands, and are useful in achieving work goals and boosting individual learning and development [22]. A previous study showed that work resources were associated with a high engagement by dentists and a feeling of enthusiasm and strength, as well as the ability to cope better with work demands [22]. Therefore, the specific needs of dentists practicing with inadequate work resources and their personal characteristics (emotional intelligence) as well as their abilities to cope with the high work demand, need to be addressed in future studies to enhance their adherence to the intervention. 

One of the strategies to cope with the issue of inadequate work resources, as suggested by many participants, was the development of a digital supporting tool to reinforce dentists’ practice of the intervention. Hence, future studies might need to explore possible solutions and utilize the co-design approach (incorporating dentists, patients, and technology experts) to conceptualize and develop useful technology for solo dentists (dentists practicing alone without an assistant), which is often the case in rural areas. The use of co-design provides substantial help for researchers, innovators, and clinicians in developing and sustaining health innovations [23].

To the authors’ knowledge, this is the first study to describe a process evaluation of a multi-component primary and secondary prevention intervention for oral cancer in Saudi Arabia. Using Linnan and Steckler’ evaluation framework, in addition to other process evaluation variables, we gained deeper knowledge on how the ISAC intervention operated in practice [11,12]. The detailed assessment of implementation through multiple data sources enhanced the validity of the study’s findings and included an observational checklist, structured questionnaires, a semi-structured interview, FGD, attendance/training logs, and the ISAC website analytics.

The challenge we faced was delivering the ISAC during COVID-19 conditions. We had a relatively small sample of dentists, due to strict COVID-19 measures imposed by the country, and this limited the number of available dentists. There might also have been a social desirability bias with participants, adopters, and implementers inclined to speak and respond positively to the intervention. We believe this bias would be of small influence, as we conducted different strategies both directly and indirectly to reduce the bias. These strategies included maintaining the participants’ anonymity, assuring confidentiality, including a statement encouraging honesty in the survey introduction, and using a 10-item form (X1) on the Marlow–Crowne scale as an independent variable in the analysis of the intervention effect [24]. The study was conducted in the Jazan region of Saudi Arabia, which may also limit the transferability of the findings to other settings without performing some adaptations. 

## 5. Conclusions

Overall, the intervention delivery attained high implementation levels for dose delivered, dose received, and fidelity. The intervention delivery was associated with high satisfaction and perceived effects. Using multiple data sources enhanced the understanding of the implementation process and strengthened the validity of the study’s findings. This elaborate process evaluation of the ISAC enabled the appraisal of implementation achievements and challenges per intervention component.

## Figures and Tables

**Figure 1 ijerph-19-07120-f001:**
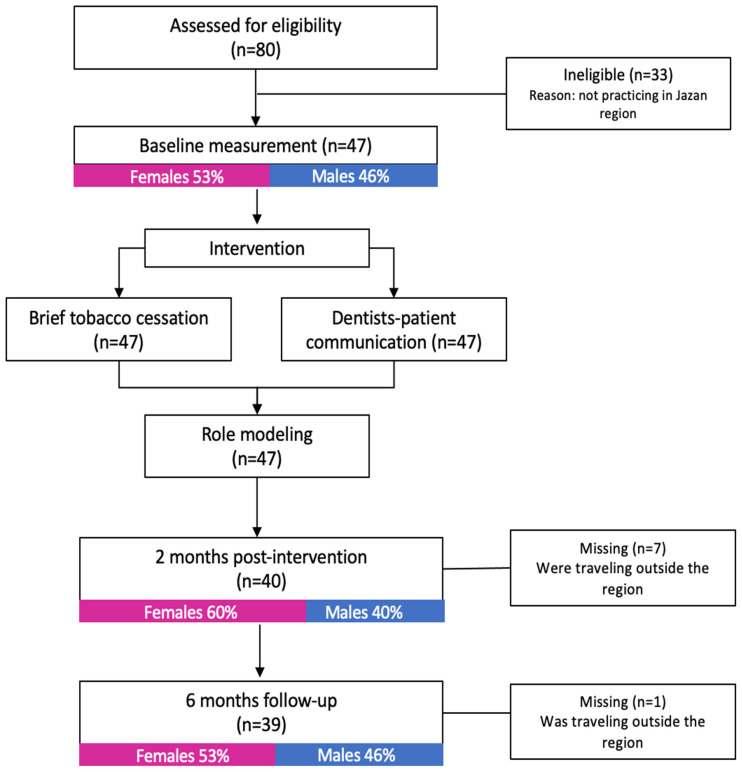
Flowchart of the process evaluation procedure.

**Table 1 ijerph-19-07120-t001:** Description of the ISAC intervention.

Session Number	Session Name	Content
Session 1.1 *n* = 47	**Tobacco-Cessation Counselling**	Didactic: discussing the importance of tobacco counseling, the role of health professionals in general and dentists in particular, the uniqueness of dental setting in tobacco-cessation activities, and an overview of the common approaches used in tobacco-cessation counseling, as well as common theoretical and evidence-based models explaining the patient current stage in the decision-making process.
Session 1.2 *n* = 47		Practical: role modeling and training of the trainers (TOT) approach to providing encouragement and capacity-building assistance for tobacco-cessation counseling.
Session 2 *n =* 47	**Patient Communication**	Role modeling and TOT by regional expert on patient-communications skills
Session 3.1 *n =* 47	**The ISAC Modeling**	Didactic: general and local OC epidemiology, general and local OC risk factors and their effects, full OC screening and referral to OC procedures (if needed), and biopsy taking, with an emphasis on patient education and practical examples, as well as introducing the ISAC method and tobacco-cessation services.
Session 3.2 *n =* 47		Practical: role modeling by the trainer (performing the ISAC), interns engaging in a role-play where they apply the ISAC on each other in two groups—the first group practices and the second group observes, and vice versa. This is followed by providing tailored feedback to each other, under the supervision of a trainer who will provide positive comments.

n = number of participating dental interns.

**Table 2 ijerph-19-07120-t002:** Perceived satisfaction, perceived benefits, and perceived difficulty of the ISAC components (*n =* 39).

Intervention Component	Satisfaction Mean (SD)	Perceived Benefits Mean (SD)	Perceived Difficulty Mean (SD)
Tobacco-cessation counseling	9.18 (0.91)	9.92 (0.27)	2.21 (1.80)
Patient communication	9.28 (0.94)	9.95 (0.22)	0.79 (0.76)
The ISAC modeling	9.77 (0.58)	9.95 (0.22)	0.51 (0.88)
Overall participation in the ISAC	9.62 (0.54)	9.72 (0.45)	1.15 (1.26)

The component rated from 0 to 10 (not at all to extremely).

**Table 3 ijerph-19-07120-t003:** Perceived effects from the ISAC participation (*n =* 39).

Participating in the ISAC Intervention Has Helped Me in:	Mean (SD)
Knowing more about oral cancer problems	3.69 (0.47)
Better understanding my role in preventing and detecting oral cancer	3.92 (0.27)
Feeling more confident in my ability to perform complete oral-cancer examinations	3.95 (0.22)
Feeling more confident in my ability to perform patient education and in tobacco-cessation counseling	3.72 (0.45)
Feeling more confident in my soft patient-communication skills	3.64 (0.48)
Performing complete oral-cancer examinations	3.95 (0.22)
Taking biopsies from a suspicious lesion	3.38 (0.63)
Educating and training patients on oral cancer self-examination	3.95 (0.22)
Providing brief tobacco-cessation counseling to my patients	3.74 (0.44)
Know-how and where I can refer patients for specialized treatments	3.79 (0.40)

Rated from 0 to 4 (strongly disagree to strongly agree).

**Table 4 ijerph-19-07120-t004:** Summary of the main findings of the ISAC process evaluation.

Intervention Implementation	Main Findings
Reach	Of the 80 dental interns, 47 (58.75%) were identified as eligible, consented, and enrolled in the intervention. All participants completed the baseline complete oral-cancer examination practice measurement baseline questionnaire (100%, *n =* 47) and received all intervention components. 85% of the enrolled dental interns completed the intervention (40) and 83% attended the six-month follow-up (*n =* 39). More females (53%) than males (46%) were recruited in the intervention at baseline and at post intervention (females = 60%, males = 40%) as well as at six-month follow-up (females = 53%, males = 46%). Of the 47 enrolled interns, 41 (87%) had returned the post-intervention questionnaire (females = 48.7%, males = 51.2%). Mean ISAC practice time was 9 ± 4 min. (range: 5 to 13 min).
Dose received	The first intervention session focused on tobacco-cessation counseling and included didactical and practical parts, using role-modeling, and training of the trainer approaches. The second session focused on enhancing dental intern’s patient-communication skills, utilizing an interactive role-modeling approach. The third session was the presentation and actual modeling of the ISAC intervention, integrating the knowledge and skills from previous sessions as well as training on other skills such as oral-cancer self-examination for patients and patient referral to specialized centers. Most participants reported reading all ISAC materials thoroughly and only a few dental interns did not read them. All enrolled dental interns visited the ISAC website to watch videos of ISAC modeling by one of their colleagues, showing patient-communications skills and the taking of intra-oral biopsies.
Dose delivered	All 47 enrolled dental interns engaged in all intervention sessions. Tobacco-cessation counseling education and training sessions took five days, four hours per day. The patient-communication interactive training session took three hours. The ISAC intervention session included a one-hour theoretical presentation, with questions and answers. The practical session of the ISAC (modeling and tailored feedback) took six hours in four successive days, with 11 to 12 dental interns per day.
Fidelity	The fidelity of the intervention sessions to the ISAC manual was high.
Satisfaction	Dental interns scored high satisfaction in all intervention components, with the highest satisfaction score for session 3; actual modeling of the ISAC (M = 9.77 ± 0.58 out of 10) Dental interns scored high in the perceived quality of information provided (M = 4.62 ± 0.63) as well as the perceived support that was received (M = 4.67 ± 0.57).
Contamination	External sources of information, including internet sites and information from other colleagues were accessed by five participants.
Context	COVID-19 national and clinical measures restricted the maximum number of eligible dental interns from 80 to 47. Dental interns had access to the ISAC materials throughout the intervention period. JDS shared a positive attitude toward practicing the ISAC and planned to institutionalize it. Dental interns were not exposed to oral cancer screening interventions other than the ISAC.

## Data Availability

The data that support the findings of this study are available from the corresponding author upon reasonable request.

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
