# Peer review of "Process Evaluation of a Comprehensive Intervention for the Early Detection and Prevention of Oral Cancer: A Mixed Methods Study"

_ijerph, 2022, doi:10.3390/ijerph19127120_

Round 1

Reviewer 1 Report

I would like to congratulate the authors for their tremendous work and the maner in which they presented the data. I agree with publication after minor spell check on the text.

Author Response

Thank you for your compliment. We thoroughly checked the spelling throughout the manuscript.

Reviewer 2 Report

Dear authors, 

This is an interesting paper explaining how training could help dentists to assess oral cancer. However, I missed a little bit more explanation in terms of statistics. How did you perform it? Please, review this part. 

1. This is an interesting exercise in order to improve oral cancer detection. The results of the study showed a satisfactory outcome from the participants. However, my question is: was there a follow-up with the participants to see if they have implemented this exercise in their current practice? Has this implementation improved the detection of oral cancer? If so, has this data compared with other data from dentists who did not attend this exercise?

2. The discussion needs a lot of work. The authors focused on pedagogical techniques, however, the main aim of this study was to assess if a new intervention would help with the early detection of oral cancer. The authors should focus the discussion in other similar studies, other interventions that were successful or not for this detection and correlate with their findings from this study. Is your intervention different than others? Did you get better/worst outcomes? Have you improved the detection of oral cancer respect to other studies?

Thank you very much. 

Author Response

Point 1: This is an interesting paper explaining how training could help dentists to assess oral cancer. However, I missed a little bit more explanation in terms of statistics. How did you perform it? Please, review this part.

Response 1: Thank you for your comment. We used descriptive statistics for all quantitative data by reporting frequency, percentage, mean ± standard deviation, which is in line with the process evaluation guideline by Moore and his colleagues. We added this in the manuscript under the data analysis section as the following: “The reach, dose delivered, dose received, recruitment, participant satisfaction and contamination were reported using descriptive statistics (frequency, percentage, mean and standard deviation)”.

Reference: Moore G F, Audrey S, Barker M, Bond L, Bonell C, Hardeman W et al. Process evaluation of complex interventions: Medical Research Council guidance BMJ 2015; 350 :h1258 doi:10.1136/bmj.h1258

Point 2: This is an interesting exercise in order to improve oral cancer detection. The results of the study showed a satisfactory outcome from the participants. However, my question is: was there a follow-up with the participants to see if they have implemented this exercise in their current practice?

Response 2: We assessed the effect of the ISAC intervention on participants’ cognition and behavior through repeated time measurements (pre-intervention, post-intervention and follow-up at six months) and reported the outcome in another study that focuses on the effect evaluation. The intervention effect was not reported in the present study because this study focuses on the process evaluation that assesses the implementation process of the complex intervention as informed by Linnan and Steckler process evaluation framework.

Point 3: Has this implementation improved the detection of oral cancer?

Response 3:  Yes. The implementation for the ISAC intervention showed positive effect on participants’ behavior toward early detection and prevention of oral cancer. The total number of reported suspicious cases of OC by dental interns had highly increased from 0 at baseline to 69 at two months post-intervention. We added this information in the manuscript within the paragraph that described the reported effect by the participants and highlighted it with yellow color (last paragraph in page 7) as the following: “In addition, the total number of reported oral potentially malignant disorders by dental interns had highly increased from zero at baseline to 69 at two months post-intervention and 85 at follow-up”

Point 4: If so, has this data compared with other data from dentists who did not attend this exercise?

Response 4: The intervention effect was not reported in the present study because this study focuses on the process evaluation that assess the implementation process of the complex intervention as informed by Linnan and Steckler process evaluation framework.

Point 5: The discussion needs a lot of work. The authors focused on pedagogical techniques, however, the main aim of this study was to assess if a new intervention would help with the early detection of oral cancer. The authors should focus the discussion in other similar studies, other interventions that were successful or not for this detection and correlate with their findings from this study. Is your intervention different than others? Did you get better/worst outcomes? Have you improved the detection of oral cancer respect to other studies? Thank you very much.

Response 5: Thank you for your comments. The objective of the present study as described in the manuscript was to evaluate the implementation of a novel intervention to enhance the early detection and prevention of oral cancer. We also rephrased the objective in the introduction section to make it more clearly as the following: “Therefore, the objective of the present study was to evaluate the implementation process of a multi-component intervention for dentists in order to increase the early detection and prevention of oral cancer by dental interns in areas with a high-risk of oral cancer.” Accordingly, the discussion part entailed discussing the main findings and comparing them in lights of other evidence considering the boundaries of the current study design. Hence, discussing the effect of the intervention was beyond the scope of the current study.

Reviewer 3 Report

 I think it’s a good idea to create a network for preventing scc of oral cavity. But Also the diagnostic work up is fundamental so a good interaction between training and for example bio endoscopy .

Author Response

Point 1: I think it’s a good idea to create a network for preventing SCC of oral cavity but Also the diagnostic work up is fundamental so a good interaction between training and for example bio endoscopy.

Response 1: Thank you for your comments. The focus of the present study was to evaluate the process of intervention implementation and not the effect of the intervention, which is reported in another manuscript. The training including cutting-edge advances in detecting oral potentially malignant disorders was covered in the intervention protocol that is reported elsewhere as it is beyond the scope of the present study.

Reference: Jafer M, Moafa I, Crutzen R and Van Den Borne B. Using Intervention Mapping to Develop ISAC, a Comprehensive Intervention for Early Detection and Prevention of Oral Cancer in Saudi Arabia. J Cancer Educ. 2022. DOI: 10.1007/s13187-022-02146-y

Reviewer 4 Report

1. Abstract is not understood and methodology is not clear. please rewrite the entire abstract.

2.  you mentioned in the abstract and in the methodology the following words: dose delivered and dose received frankly writing i did not understand what kind of intervention was used.

3. please can you justify why you did not mention the importance of the  applications of salivary metabolomics in biomarker discovery in oral cancers.

4. why you did not use in your study the technique of using the oral gargle to detect the Methylation of EPB41L3 tumor suppressor gene

5. Methodology section need to be rephrasing and providing further expination in order to be understood

6. Conclusion section is very confusing. please rewrite it again

Author Response

Point 1: The abstract is not understood and methodology is not clear. Please rewrite the entire abstract.

Response 1: Thank you for your comments. We considered your suggestion and restructured the abstract as the following (yellow highlighted in the manuscript):

“Process evaluations helps to understand and refine health interventions. The “ISAC” intervention targeted enhancing early detection and prevention of oral cancer (OC) through tobacco cessation counseling, patient communications and ISAC role-modeling. Linnan and stickler process evaluation’s framework specified indicators which were assessed throughout ISAC implementation by mixed methods on context, reach, dose delivered, dose received, fidelity, recruitment and satisfaction from participants in the Jazan region of Saudi Arabia over six months. Findings showed that 47 of 80 (58.75%) eligible dentists were reached and received all components. Thirty-six (76.6%) participants reported reading all intervention materials, visited the ISAC website and scored high in the perceived quality of provided information (M=4.62 ± 0.63) and provided support (M=4.67 ± 0.57). The fidelity was high across all intervention components. Role-modeling of the ISAC had the highest satisfaction score (M=9.77 ± 0.58 out of 10). High perceived-effects were re-ported in relation to feeling confidence, performing OC examination and training patients on OC self-examination (3.95±0.22). The intervention attained high implementation levels for dose delivered, dose received, and fidelity. The intervention delivery was associated with high satisfaction and perceived effects. Using multiple data sources enhanced the understanding of the implementation process and strengthened the validity of the study’s findings.”

Point 2: You mentioned in the abstract and in the methodology the following words: dose delivered and dose received frankly writing I did not understand what kind of intervention was used.

Response 2: Dose delivered is the amount of the intervention that was delivered to the participants. While dose received is the extent of intervention content that was received by participants. The used intervention and its components were specified in table 1 in the manuscript (also highlighted in yellow). It is a comprehensive cognitive and behavioral intervention guided by the intervention mapping approach. Detailed information on the intervention can be found in the published protocol that is also specified in the present study.

Reference: Jafer M, Moafa I, Crutzen R and Van Den Borne B. Using Intervention Mapping to Develop ISAC, a Comprehensive Intervention for Early Detection and Prevention of Oral Cancer in Saudi Arabia. J Cancer Educ. 2022. DOI: 10.1007/s13187-022-02146-y

Point 3: Please can you justify why you did not mention the importance of the applications of salivary metabolomics in biomarker discovery in oral cancers. Why you did not use in your study the technique of using the oral gargle to detect the Methylation of EPB41L3 tumor suppressor gene.

Response 3: We agree on the importance of the salivary metabolomics in biomarker discovery of oral cancer and the use of oral gargle to detect the Methylation of EPB41L3 tumor suppressor gene. These types of tests are not available in the study context clinics which could be a weakness point. However, upon the needs assessment of the oral cancer delayed detection issue in the Jazan region, we found that the problem mainly linked to the behavioral aspects of dentists; dentists were not practicing comprehensive oral cancer screening, they lacked confidence on their ability to perform screening, they believed other dentists not practicing oral cancer screening nor patient education of oral cancer risk factors. Therefore, we focused on the changeable factors, strengths, and what is available.

The specification of the salivary metabolomics in biomarker discovery and other techniques will be detailed within the effect evaluation manuscript (to be published soon). In general, the instructions and the training that were given to dentists regarding the screening for oral cancer followed the oral cancer clinical practice guideline including, but not limited to, taking patient history (general history and oral history), performing visual examination of extraoral tissues (head, neck, lymph nodes, lips and perioral tissues), performing intraoral examination (inspect and palpate soft tissues with more attention to high risk areas such as below and both sides of the tongue, floor of the mouth and soft palate), evaluating the characteristic of the lesion (color, size, texture and outline) and the use of adjunctive technique such as toluidine blue staining, direct fluorescence visualization as well as taking tissue biopsy. We also added this part to the manuscript and highlighted them in yellow color.

References: Jafer, M, Crutzen, R, Halboub, E, Moafa I. Dentists Behavioral Factors Influencing Early Detection of Oral Cancer: Direct Clinical Observational Study. J Canc Educ. 2020. https://doi.org/10.1007/s13187-020-01903-1

Jafer M, Crutzen R, Ibrahim A, Moafa I, Zaylaee H, Ajeely M et al. Using the Exploratory Sequential Mixed Methods De-sign to Investigate Dental Patients' Perceptions and Needs Concerning Oral Cancer Information, Examination, Prevention and Behavior. Int J Environ Res Public Health. 2021;18(14):7562. doi:10.3390/ijerph18147562

Oral Cancer Clinical Practice Guideline - CDSBC. Guideline for the Early Detection of Oral Cancer in British Columbia, 2008, https://www.cdsbc.org/CDSBCPublicLibrary/Oral-Cancer-Clinical-Practice-Guideline.pdf.

Point 4: Methodology section need to be rephrasing and providing further explanation in order to be understood.

Response 4: Thank you for your comment. We considered your comments, rephrased the methodology section, and provided further explanation to ease understanding. (Yellow highlighted in text)

Point 5: Conclusion section is very confusing. Please rewrite it again.

Response 5: Thank you for your suggestion. We considered your suggestion and rephrased the conclusion section as the following (Yellow highlighted in text): “Overall, the intervention delivery attained high implementation levels for dose delivered, dose received, and fidelity. The intervention delivery was associated with high satisfaction and perceived effects. Using multiple data sources enhanced the understanding of the implementation process and strengthened the validity of the study’s findings. This elaborate process evaluation of the ISAC intervention enabled the appraisal of implementation achievements and challenges per intervention component”.

Round 2

Reviewer 4 Report

it seemed to be OK.